# Assessing the Larvicidal Properties of Endemic Campeche, Mexico Plant *Piper cordoncillo* var. *apazoteanum* (Piperaceae) against *Aedes aegypti* (Diptera: Culicidae) Mosquitoes

**DOI:** 10.3390/insects14040312

**Published:** 2023-03-24

**Authors:** Nancy Alonso-Hernández, Carlos Granados-Echegoyen, Ileana Vera-Reyes, Rafael Pérez-Pacheco, Fabián Arroyo-Balán, Alejandro Valdez-Calderón, Arián Espinosa-Roa, Henry Jesús Loeza-Concha, Evert Villanueva-Sánchez, Florinda García-Pérez, Fidel Diego-Nava

**Affiliations:** 1Interdisciplinary Research Center for Integral Regional Development (CIIDIR), Oaxaca Campus, National Polytechnic Institute (IPN), Hornos 1003, Oaxaca 71230, Mexico; alonsoh_nancy@hotmail.com (N.A.-H.); fdiego@ipn.mx (F.D.-N.); 2Applied Entomology Laboratory, Center for Studies in Sustainable Development and Wildlife Use (CEDESU), CONACYT-Autonomous University of Campeche, Campeche 24079, Mexico; fabian.arroyo@conacyt.mx; 3Biosciences and Agrotechnology Department, CONACYT-Center for Research in Applied Chemistry, Saltillo 25294, Mexico; 4Technological University of the Metropolitan Zone of the Valley of Mexico, Hidalgo 43816, Mexico; a.valdez@utvam.edu.mx; 5Innovation and Technological Research Park (PIIT), CONACYT-Center for Research in Applied Chemistry, Monterrey 66628, Mexico; arian.espinosa@ciqa.edu.mx; 6Postgraduate College Campus Campeche, Campeche 24450, Mexico; loeza.jesus@colpos.mx; 7National Laboratory for Agrifood and Forestry Research and Service, CONACYT-University of Chapingo, Chapingo 56230, Mexico; evillanueva@conacyt.mx; 8Agronomy Department, Novauniversitas, Oaxaca 71513, Mexico; florgp@novauniversitas.edu.mx

**Keywords:** Piperaceae, endemic plant, Campeche, mosquito, growth inhibition

## Abstract

**Simple Summary:**

Mosquitoes can significantly damage to human health by transmitting a wide range of diseases, including malaria, dengue fever, Zika virus, and West Nile virus. These diseases can result in illness, hospitalization, and death. In addition, mosquito bites can cause itching, swelling, and irritation, leading to discomfort and a loss of productivity. In recent years, plant-derived products have gained increased attention as alternative options to synthetic pesticides. Secondary metabolites from endemic plants have been shown to possess unique chemical structures and diverse biological activities, making them valuable resources. Some well-known secondary metabolites from endemic plants include alkaloids, flavonoids, terpenoids, and phenolic compounds. These plant-derived products are often biodegradable, have low toxicity to non-target organisms, and can be produced sustainably. Extracting metabolites from endemic plants has several uses, including the development of new insecticides products.

**Abstract:**

The research aims to investigate the mortality effect of essential oil from *Piper cordoncillo* var. *apazoteanum*, an endemic plant from Campeche, Mexico, on early second-instar *Aedes aegypti* larvae; it also aims to identify the volatile compounds present in the fresh leaves of the plant. To test the effectiveness of the essential oil, we followed World Health Organization Standard Procedures. Larvae were observed for 17 consecutive days after treatment to determine the mortality and growth-inhibitory effect exerted by the essential oil. The results showed that the essential oil was effective in controlling mosquito populations. At a concentration of 800 ppm, the oil achieved an effectiveness rate of 70.00 ± 8.16% after 24 h, increasing to 100.00 ± 0.01% mortality after 72 h. With a concentration of 400 ppm, the effectiveness was 98.33 ± 0.17% by the end of the experiment. Furthermore, the obtained results demonstrated that the LC_50_ value was 61.84 ± 6.79 ppm, while the LC_90_ value was 167.20 ± 11.49 ppm. Essential oil concentrations inhibited the growth of immature insect stages, with concentrations between 800–100 ppm demonstrating very high inhibitory activity, and the lowest concentration of 50 ppm showing high inhibitory activity. The study also identified 24 chemical compounds representing 86.71% of the volatile compound composition of the fresh leaves of *P. cordoncillo*; the most abundant compounds were Safrole, Caryophyllene oxide, E-Nerolidol, and Calarene epoxide. The method used to extract the volatile compounds, solvent-free microwave extraction (SFME), is a promising alternative to traditional methods that avoids the use of potentially harmful solvents, making it more ecologically friendly and potentially safer for professionals handling the extracted compounds. Overall, the study demonstrates the potential of *P. cordoncillo* essential oil as an effective means of controlling mosquito populations, and provides valuable information on the chemical composition of the plant.Moreover, our study is the first to report on the biological activity and chemical composition of *P. cordoncillo* worldwide.

## 1. Introduction

Diseases transmitted by mosquitoes, such as *Aedes aegypti* (Diptera: Culicidae), pose a significant public health challenge globally, with an estimated 390 million dengue infections occurring each year [1]. In Mexico, mosquito-borne diseases continue to be a major concern, with dengue, chikungunya, and Zika being the most commonly reported diseases [2]. *Aedes aegypti* is the primary vector responsible for transmitting these diseases; traditional control methods, such as insecticide spraying, have been largely ineffective in controlling mosquito populations. Therefore, there is a need to explore new and effective methods for controlling mosquito populations [3,4].

One promising approach to mosquito control is the use of essential oils extracted from plants of the Piperaceae family, a large family of plants that includes more than 3600 species, many of which are known for their medicinal and insecticidal properties [5]. The secondary metabolites produced by Piperaceae plants have been shown to possess a range of biological activities, including insecticidal, antifungal, and antioxidant properties [6,7,8].

In recent years, there has been growing interest in the use of essential oils extracted from Piperaceae plants for mosquito control [9,10,11]. Several studies have demonstrated that essential oils from Piperaceae plants have potent insecticidal and repellent properties against *Ae. aegypti* larvae and adults [12,13,14]. For example, Santana et al. [15] found that essential oils from *Piper marginatum* and *Piper aduncum* (Piperaceae) had significant larvicidal activity against *Ae. aegypti*, with lethal concentration LC_50_ values of 34.0 and 46.0 ppm, respectively; Silva et al. [16] also mentioned that species such as *Piper hispidinervum* can be applied as sustainable control tools for this mosquito. Similarly, Kamaraj et al. [17] reported that *Piper nigrum*-derived natural products had strong repellent and insecticidal activities [18]. Beyond their insecticidal properties, essential oils extracted from Piperaceae plants have several advantages over traditional insecticides. For example, they are generally considered safe for human and animal use, biodegradable, and have lower toxicity to non-target organisms; these factors make them more environmentally friendly than many synthetic insecticides [19,20].

Furthermore, essential oils have a complex mixture of compounds, which may make it more difficult for insects to develop resistance [21,22,23]. Essential oils are composed of various chemical compounds, including monoterpenes, sesquiterpenes, and phenylpropanoids; these compounds are responsible for essential oils’ insecticidal activity [24].

*Piper cordoncillo* var. *apazoteanum* is a plant species endemic to Campeche, Mexico [25]. This plant is commonly known as “Momo” or “apazote” in the region [26]. It is a shrub that can grow up to 5 m in height, with long leaves and small, greenish-white flowers [27]. The plant has traditionally been exploited in the region for its medicinal properties, specifically for the treatment of gastrointestinal; it is also used in traditional cuisine. There are currently no reports of its biological activity on insect pests or chemical composition. Therefore, our work is the first report on this topic.

Several chemical compounds have been identified in various parts of the Piperaceae family. One of the main compounds found is piperine, a bioactive alkaloid responsible for the plant’s pungent taste and medicinal properties. Piperine has been found to have anti-inflammatory, antioxidant, and anticancer activities [28,29,30]. Other chemical compounds found in essential oils include flavonoids and terpenes; these compounds have anti-inflammatory, antidiabetic, antioxidant, antifungal, and insecticidal activities [31,32,33,34]. The essential oils extracted from these plants contain compounds such as β-pinene, limonene, and caryophyllene, which are known for their biological activities [35,36].

Despite these promising results, the use of essential oils for mosquito control in Mexico is still in its early stages. The two objectives of this study are: to investigate the insecticidal potential of essential oils derived from the endemic plant species from Campeche, Mexico, for specific use in mosquito control, and; to identify the chemical compounds responsible for the insecticidal activity of essential oils. The unique characteristics of the “Momo” plant make it a promising resource for the development of natural products for insect control, which can contribute to the preservation of biodiversity, and the development of sustainable and environmentally friendly mosquito control strategies in the region.

## 2. Materials and Methods

The study was carried out at the Laboratory of Aquatic Ecology and Environmental Monitoring of the Center for Sustainable Development Studies and Wildlife Exploitation (CEDESU), located at the Autonomous University of Campeche (19°48′5.09″ N, 90°30′17.265″ W); the analysis of the essential oil occurred at the Technological University of the Metropolitan Zone of the Valley of Mexico.

### 2.1. Plant Collection and Extraction of Essential Oil

Fresh leaves of the “Momo” plant were collected from the wild in the Lázaro Cárdenas neighborhood of San Francisco de Campeche, Campeche, Mexico (19°48′28.7″ N, 90°32′06.7″ W). The samples were transported to the laboratory, where they were washed with distilled water to remove impurities and stored in refrigeration (4 °C) until used. The plant underwent authentication procedures under the supervision of Msc. Ricardo Gongora-Chin, the curator of the CEDESU–UAC herbarium. To document the plant’s identity, a sample was acquired and added to the herbarium’s collection; it was given the voucher number BUAC-#0321 as its official reference code. At the time of use, maceration was carried out at a ratio of 1:10 (*w*/*v*), using 100 g of fresh plant material and 1000 mL of water. The mixture was left to hydrate for 30 min and then drained in the shade for 5 min. This operation was repeated 10 times separately. For the oil extraction, a solvent-free microwave extraction (SFME) NEOS–GR System (Milestone, Italy) was used, adjusting the system between 300 and 600 W for irradiation, with three repetitions of 15 min and rests of 5 min for 500 g of the plant; this operation was carried out three times until enough essential oil was obtained for the bioassays and chemical analysis. The resulting essential oil was stored in a dark, sealed container at 4 °C until used [37,38,39].

### 2.2. Identification of Volatile Compounds

The essential oil of *P. cordoncillo* was analyzed by gas chromatography-mass spectrometry (GC-MS), using a Thermo Scientific GC TRACE 1310 EM ISQ LT apparatus; the apparatus was operated in Electron Ionization mode (70 eV), equipped with split/splitless injector (250 °C), and used a TG–SQC Thermo Scientific capillary column [15 m × 0.25 mm (i.d.), film thickness: 0.25 μm]. The temperature for the TG–SQC column was 50 °C (5 min) to 250 °C at a rate of 20 °C/min. Helium was used as a carrier gas at a flow rate of 1 mL/min. The identification of the components was based on a comparison of their mass spectra with those reported in the database NIST MS Search 2.0 (National Institute of Standards and Technology Mass Spectral Database), and/or by comparison of their relative retention index (RRI) to a series of n-alkanes. Alkanes were used as reference points in the calculation of relative retention indices (RRI). Relative percentage amounts of the identified components were calculated from FID chromatograms [40,41].

### 2.3. Mosquito Breeding

*Aedes aegypti* mosquitoes were bred in the laboratory by collecting larvae and pupae from stagnant, contaminated water near San Francisco de Campeche. Larvae were fed tilapia fish food (Api-tilapia-1^®^) until they reached the pupal stage, at which point they were transferred to containers (47 cm × 35 cm × 12 cm) with water, and introduced into insect cages for development into adults. Adult mosquitoes were provided with a 10% sugary solution; a sedated rabbit (*Oryctolagus cuniculus*) was provided as a blood meal for the female mosquitoes. Black plastic containers (30 × 20 × 6 cm) were placed in the insect cages, with Pellon^®^ fabric attached to the walls on which gravid females could lay eggs; the resulting egg-laying material was collected and dried for later use. The breeding environment was maintained at a temperature of 27 ± 2 °C, 60–70% relative humidity, and a 12-h light/dark photoperiod. Taxonomic keys were used to identify the mosquitoes at the genus and species level. The resulting F1 generation of mosquitoes was used for a bioassay [42,43].

### 2.4. Preparation of Test Concentrations

From the essential oil stock solution, 0.8 mL of oil was solubilized in 10 mL of distilled water with 0.001% polysorbate (Tween-20) as an emulsifying agent to prepare test concentrations. Concentrations of 50, 100, 200, 400, and 800 ppm were assigned in quadruplicate for each of the 80 larvae tested [44].

### 2.5. Larvicidal Bioassay

To investigate the mortality effect on early second-instar *Ae. aegypti* larvae, the study evaluated their development over three consecutive days and for 17 days after treatment application. The World Health Organization Standard Procedures, with slight modifications, were followed to observe the different developmental stages (second, third, and fourth instar, as well as pupae, and adult) [45]. The experimental assays were set up, with 20 larvae placed in plastic cups containing 99 mL of distilled water and 1 mL of essential oil. Mortality was assessed using two criteria: (1) when a larva did not exhibit movements similar to those of the Control Group, and; (2) when a larva did not show any reaction after being disturbed with a brush in the siphon of its cervical region. Each treatment was replicated four times.

### 2.6. Growth Inhibition

Early second instar larvae were used in the bioassay, as they are more sensitive and vulnerable to experimental treatments, which allows a more precise and reliable evaluation of the inhibitory effect of the substances under study. Additionally, this instar was selected because the larvae are more uniform in terms of size and age, which minimizes variability in the results. Each experimental unit consisted of a 125 mL plastic cup, filled with 99 mL of distilled water and 20 larvae. Each experimental unit received 1 mL of the concentrations used. Each bioassay included a control without treatment application. When the control without application formed between 90–93% pupae, the number of live and dead organisms (larvae and pupae), and the number of emerged adults, were recorded. An adult was considered dead if it remained trapped in the pupal exuvia, while a dead larva or pupa exhibited abnormal movements when disturbed with a dissection needle, using the methodology of Granados–Echegoyen et al. [46]. With the collected information, the relative growth inhibition (RGI) was quantified using the formula of Zhang et al. [47]:$$\mathrm{RGI}=\frac{\sum_{1}^{4}\left(\mathrm{No}.\;\mathrm{of}\;\mathrm{live}\;\mathrm{insects}*\mathrm{insect}\;\mathrm{stage}\right)+\sum_{1}^{4}\left[\mathrm{No}.\;\mathrm{of}\;\mathrm{dead}\;\mathrm{insects}*\left(\mathrm{insect}\;\mathrm{stage}-1\right)\right]}{\left(\mathrm{No}.\;\mathrm{total}\;\mathrm{of}\;\mathrm{insects}\;\mathrm{evaluated}*\mathrm{total}\;\mathrm{phases}\;\mathrm{of}\;\mathrm{the}\;\mathrm{insect}\right)}$$
where 1, 2, 3, and 4 correspond to the second, third, and fourth instar, and the pupal stages of the insect, respectively. The number of insects used per concentration was 80, with a total number of immature insect stages of four (second, third, and fourth instar larvae, and one pupa).

The RGI data of mosquitoes treated with the essential oil of *P. cordoncillo* were grouped into categories proposed by Arivoli et al. [48] with some modifications, as follows:

- → No inhibitory growth activity (RGI ≥ 1.00)

+ → Low inhibitory activity (0.75 ≥ RGI ≥ 0.99)

++ → Moderate inhibitory activity (0.50 ≥ RGI ≥ 0.74)

+++ → High inhibitory growth activity (0.25 ≥ RGI ≥ 0.49)

++++ → Very high inhibitory growth activity (0.00 ≥ RGI ≥ 0.24)

The Control Group, which contained 80 larvae, was treated with 400 mL of distilled water in place of the essential oil.

### 2.7. Statistical Analysis and Experimental Design

The bioassays were conducted individually under a completely randomized experimental design for each variable studied. The data were tested for normality of errors using the Shapiro-Wilk test, and for homogeneity of variances using the Bartlett test. One-way analysis of variance (ANOVA) and comparison of means were performed using Tukey’s test (with a significance level of *p* < 0.05) as a post hoc test, which was carried out using Minitab version 18.1. Mortality data from each concentration assay underwent Probit analysis to estimate the LC_50_ and LC_90_ values [49]. The data presented in tables shows the means and standard deviations for the mortality and relative growth index variables; the lethal concentrations are expressed as means and standard errors of the mean with 95% confidence intervals, allowing for a range of values that meet reliability standards.

## 3. Results

### 3.1. Identification of Volatile Compounds

The volatile compounds obtained through solvent-free microwave extraction (SFME, method for extracting compounds from plants without using solvents) of fresh leaves of *P. cordoncillo* (Figure 1) are shown in Table 1. By comparing the mass spectra of each compound with the data stored in the NIST MS Search 2.0 (database used to compare mass spectra of chemical compounds and identify them based on their spectral patterns), 24 chemical compounds representing 86.71% of their composition were identified. The most abundant compounds were Safrole (20.67%), Caryophyllene oxide (13.24%), *E*-Nerolidol (9.56%), and Calarene epoxide (5.04%) (Figure 2). Among the least present compounds, we have trans-longipinocarveol (4.61%), Myristicin (4.30%), allo-aromadendrene epoxide (3.87), Cubelol (2.53%), and Aromadendrene oxide-(2) (2.17%); the remaining compounds represent 20.75% of the total, and their concentrations range between 1.98% and 0.78% (Figure 3 and Figure 4).

### 3.2. Larvicidal Action and Growth Inhibition

The results of larvicidal action are particularly promising, as they suggest that the essential oil treatment could be an effective means of controlling mosquito populations. Furthermore, the fact that efficacy increased over time suggests that repeated applications of the treatment could result in even greater mosquito mortality rates. Significant differences were observed over the three-day monitoring period, with an increase in efficacy as the immature mosquitoes were exposed to the essential oil treatment. At a concentration of 800 ppm, the oil achieved an effectiveness rate of 70.00 ± 8.16% after 24 h of treatment (*p* < 0.001, F = 36.89, df = 5,18, χ² = 91.11), increasing to 100.00 ± 0.01% mortality after 72 h (*p* < 0.001, F = 181.56, df = 5,18, χ² = 98.01) (Table 2). However, further research will be needed to confirm these findings, and to investigate the potential side effects of the essential oil treatment on non-target species.

The experiment was concluded once the Control Group showed a 90–93% rate of pupal formation; it was then found that more larvae in the second stage died compared to those in the third and fourth stages. After 17 days of treatment application, the total mortality of the immature insect stages was quantified; these data show that the concentration of 400 ppm achieved an effectiveness of 98.75 ± 2.50%, whereas the lowest concentration of 50 ppm achieved a 60.30 ± 20.60% effectiveness (*p* < 0.001, F = 45.25, df = 5,18, χ² = 92.63). According to the categories proposed by Arivoli et al. [48], all the evaluated concentrations inhibited the growth of immature insect stages, concentrations of 800–100 ppm demonstrate very high inhibitory activity, while the lowest concentration of 50 ppm show high inhibitory activity (*p* < 0.001, F = 59.65, df = 5,18, χ² = 94.31) (Table 3). These findings suggest that the treatments were effective in hindering the growth and development of the studied insect’s immature stages, with the highest efficacy observed at concentrations of 800–100 ppm. This indicates that these concentrations could potentially be utilized for effective insect population control.

The results showed that the LC_50_ values required for controlling 50% of the population at 48 and 72 h were 230.30 ± 19.05 ppm and 146.30 ± 12.62 ppm, respectively. Furthermore, the study was extended for 17 days, which enabled the researchers to determine the LC_50_ and LC_90_ values required for effective insect control. The obtained results demonstrated that the LC_50_ value was 61.84 ± 6.79 ppm, while the LC_90_ value was 167.20 ± 11.49 ppm (Table 4).

Substances that inhibit the growth of mosquito larvae prevent them from developing into adults. This means that the larvae do not reach the adult stage of their life cycle, and instead remain in the larval stage. As a result, the larvae can become malformed or weakened, as they are not able to fully develop into strong and healthy adults. If the larvae are unable to successfully complete their life cycle, they may also fail to develop into pupae, or may develop into weak and unhealthy pupae that are unlikely to survive to adulthood. In some cases, adults may emerge from the pupae but be weakened and unable to fly or reproduce, leading to a gradual decline in the mosquito population. Ultimately, the use of substances that inhibit the growth of mosquito larvae can reduce the overall number of adult mosquitoes, and help to prevent the spread of mosquito-borne diseases.

## 4. Discussion

The use of solvent-free microwave extraction (SFME) represents a promising alternative to traditional methods for extracting volatile compounds from plants. This method is attractive because it avoids the use of potentially harmful solvents, making it eco-friendlier and potentially safer for professionals handling the extracted compounds. The results of the current study provide valuable insights into the volatile compounds present in the fresh leaves of *P. cordoncillo*. The identification of 24 chemical compounds, representing 86.71% of the total composition, is a significant achievement that could have important implications for the potential uses of this plant. Of the identified compounds, Safrole, Caryophyllene oxide, E-Nerolidol, and Calarene epoxide were found to be the most abundant. These compounds are known to have a range of biological activities, including antimicrobial, antifungal, and insecticidal properties [60,61,62]. Their presence in *P. cordoncillo* suggests that this plant may have potential applications in the development of new insecticidal agents. It is also worth noting that the least present compounds, including trans-longipinocarveol, myristicin, and allo-aromadendrene epoxide, may also have important biological activities, despite their lower concentrations [63,64]. Further research will be needed to fully understand the potential uses of these compounds, and to explore the optimal methods for extracting them from *P. cordoncillo*.

The most studied characteristic of essential oils is their antioxidant capacity, which acts as a defense mechanism in the unsaturation of lipids in animal tissue and is associated with a hepatoprotective effect in mammals. The effect of oxygen in humans is of utmost importance, as it can produce reactive oxygen species (ROS) that can be harmful to cells and tissues. Essential oils work preventively by reducing the activation of ROS and increasing their detoxification [65,66]. Regarding the larvicidal activity of essential oils, one proposed mechanism is the inhibition of trypsin in the intestines of larvae. Trypsin is a serine protease that is widely found in the intestines of insects; impairment of its activity can result in poor nutrient absorption and the unavailability of essential amino acids. Other potential targets of essential oils include transient receptor potential (TRP)-like ion channels, acetylcholinesterase, tyramine, octopamine, and gamma amino butyric acid (GABA) receptors [67,68]. In the case of the oils characterized in our extract, the largest safrole constituent has been reported as the most toxic terpene of *Asarum heterotropoides* (Aristolochiaceae) and *P. betle* against the *Culex pipiens pallens*, *Ae. aegypti*, and *Ochlerotatus togoi* (Culicidae) mosquito species [69,70].

Essential oils from this plant species exhibit a complex diversity of chemical elements. These secondary metabolites have long been used empirically to control vectors and causative agents of diseases. Therefore, it is necessary to continue studying the extensive diversity of endemic flora present in certain regions.

The growth-regulating effect of compounds that mimic juvenile hormones in arthropods is a well-documented phenomenon in the literature. These compounds can be found in a variety of plant species, including those that are commonly used for their essential oils. Juvenile hormone is a key hormone that regulates the growth and development of insects, including their transition from larval to pupal, and then to adult, stages. It is involved in the regulation of various physiological processes, including molting, metamorphosis, reproduction, and behavior. The regulation of juvenile hormone levels is critical for proper insect development and survival. Compounds that mimic juvenile hormones are often referred to as juvenile hormone analogs (JHAs) or juvenile hormone mimics (JHMs). These compounds have been shown to interfere with normal juvenile hormone signaling, and can disrupt normal insect development. They can also cause malformations in developing insects and ultimately lead to mortality. These compounds have been found to have a range of effects on different insect species, including mosquitoes, flies, and beetles [71,72,73,74,75].

The Piperaceae family contains chemical compounds with the potential for mosquito control, enabling them to act against mosquitoes and other pest insects. For example, plant species from other botanical families, even from the same genus but different species, have been investigated. Similar secondary metabolites to those found in our study have been reported, such as the research conducted by Leyva et al. [76]; their research reports safrole as the major compound in the essential oil of *P. auritum*, with a possible presence of 93.24%, and the chemical compound myristicin in a smaller amount of 4.34%. The authors calculated the median lethal concentration (LC_50_) of 17 mgL^−1^ against larvae of the mosquito *Ae. aegypti.* Additionally, reports indicate that the compound concentration may vary according to the season in which the plant is collected, as shown in the study conducted by Ortiz et al. [77]. They recorded a variation in the concentration of safrole found in the essential oil of *Laureliopsis philippiana* (Monimiaceae) of 38.47% in autumn, 59.11% in winter, 64.7% in spring, and 57.98% in summer. They found that the repelling effect on the maize weevil *Sitophilus zeamais* (Coleoptera: Curculionidae) increased proportionally with the compound concentration in the essential oil. The authors attribute the plant’s effectiveness to the high content of safrole potentiated by the presence of linalool.

Moreover, Descamps et al. [78] reported that the essential oil extracted from the leaves and fruits of *Schinus areira* (Anacardiaceae) has a repellent and lethal effect on larvae and adults of the red flour beetle *Tribolium castaneum* (Coleoptera: Tenebrionidae). They found 3-carene, a bicyclic monoterpene with a unique propane ring, at a concentration of 20.8% in the essential oil of this plant. However, this compound has not only shown effectiveness against pest insects, but also against fungi, such as *Rhizoctonia solani* (Agonomicetaceae), *Pythium irregulare* (Pythiaceae), *Ceratocystis pilifera* (Ceratocystidaceae), *Phragmidium violaceum* (Phragmidiaceae), and *Fusarium oxysporum* (Nectriaceae). These were exposed to dilutions of 30% and 50%, as well as undiluted (100%) essential oils of *Peumus boldus* and *Laureliopsis philippiana* (Monimiaceae), showing favorable fungistatic activity. Terpenes such as 3-carene, α-felandrene, and α-pinene [79], were found in the essential oil, similar to those in the previous study.

Furthermore, Fenchone, an organic compound classified as a monoterpenoid present in the essential oil of *Plectranthus incanus* (Lamiaceae) at a concentration of 6%, has been reported to have a repelling effect of 100% against mosquitoes *Anopheles stephensi* and *C. fatigans* when applied at concentrations of 10 μL cm^−2^. It provides over 300 min (5 h) of protection against mosquito bites [80].

The compounds identified in our study belong to chemical groups such as flavonoids and terpenes. Some terpenes can be toxic to insects, causing damage to their nervous system, digestive system, or other physiological processes. Terpenes can also interfere with the behavior of insects [81,82,83]. Flavonoids can affect the insect’s digestive system by inhibiting the activity of digestive enzymes, disrupting nutrient absorption, and causing starvation. They can prevent the breakdown of carbohydrates and proteins in insects, leading to reduced growth and reproduction [84,85,86]. Terpenoids are a diverse class of organic compounds that are often responsible for the characteristic odor and flavor of plants [87]. They can act on insects by disrupting their cellular membranes or interfering with their hormone systems, leading to dehydration and death. They can even interfere with the molting process of insects, leading to reduced growth and development [88,89]. However, it should be understood that the complex mixture of chemical substances present in essential oils may work synergistically, individually, or even be ineffective [90,91].

The major compounds found in our study have reports of biological activity on pests; examples include safrole and myristicin, which are natural compounds found in several plant species. Both compounds have been studied for their potential insecticidal properties, particularly against mosquito larvae. In one study, safrole was shown to have significant larvicidal activity against *Ae. aegypti*, the mosquito that transmits the dengue, chikungunya, and Zika viruses. Myristicin has also been shown to have larvicidal activity against mosquito species, including *Ae. albopictus* and *Cx. pipiens* [92,93,94]. Another compound, *E*-Nerolidol, has been studied for its potential insecticidal and repellent properties against a variety of arthropods, including mosquitoes, ticks, and agricultural pests [95,96,97,98]. Additionally, Caryophyllene oxide has been studied for its potential effects against termites and grain weevils [99,100,101]. The potential insecticidal properties of Calarene have been studied against the stored product pest *Sitophilus zeamais* [102].

It’s important to note that insecticidal compounds have different modes of action on insects, and their efficacy and safety can vary depending on the pest species, concentration, and application method. The optimization process in essential oil extraction is an important step in producing high-quality, valuable essential oils. By carefully selecting plant material, choosing the right extraction method, and optimizing operating conditions, it is possible to maximize yield and produce a high-quality product that meets the needs for pest control. Solvent-free microwave extraction (SFME) is a relatively new extraction technique that uses microwave radiation to extract organic compounds from a sample material without the use of a solvent. The problems associated with traditional extraction methods, such as the use of large amounts of organic solvents, long extraction times, and low yields, have been addressed with SFME. One of the main advantages of SFME is that it is a eco-friendly and sustainable extraction technique that does not use any organic solvents. This reduces the environmental impact of the extraction process, and eliminates the need for solvent disposal. SFME is also a fast extraction technique; it can extract compounds in a matter of minutes, whereas traditional extraction methods can take hours or even days [103,104,105].

It is important to mention the information provided by Nicoletti [106], who emphasizes the need for a collaborative, multidisciplinary approach to combat insect-borne diseases, which pose a significant global health concern. The author advocates for the use of natural products, such as essential oils, as alternative insecticides in combination with synthetic insecticides to reduce the development of insect resistance. Furthermore, the author highlights the importance of further research and development in this field to identify new strategies for vector control and disease treatment.

Similarly, the comprehensive review conducted by Salehi et al. [107] on the chemical compounds and biological activities of Piperaceae species provides a valuable foundation for our study. By focusing on the phytochemistry, biological activities, and potential applications of the Piper species in our study, we aim to contribute to the scientific literature on this taxonomic family and its members. Our research may be of interest to those working in the field of natural products research.

## 5. Conclusions

This is the first study to document the chemical composition and biological effects of *Piper cordoncillo var. apazoteanum*, a plant species endemic to Campeche, Mexico. Using solvent-free microwave extraction (SFME), the researchers identified 24 chemical compounds representing 86.71% of the plant’s volatile compound composition. The most abundant compounds were Safrole, Caryophyllene oxide, E-Nerolidol, and Calarene epoxide. Results showed that *P. cordoncillo* has potential applications in the development of new insecticidal agents, while the larvicidal action of the plant’s essential oil treatment showed promising results in controlling mosquito populations. Higher concentrations of the plant’s compounds had a more significant inhibitory effect on the growth and development of immature insect stages; concentrations between 800–100 ppm could potentially be utilized for effective insect population control. The study also determined the LC_50_ and LC_90_ values required for effective insect control to be 61.84 ± 6.79 ppm and 167.20 ± 11.49 ppm, respectively. Further investigation is suggested to explore the potential applications of both polar and non-polar fractions of this plant, as well as to examine its effects on other organisms, toxicological implications, and effects on non-target insects.

## Figures and Tables

**Figure 1 insects-14-00312-f001:**
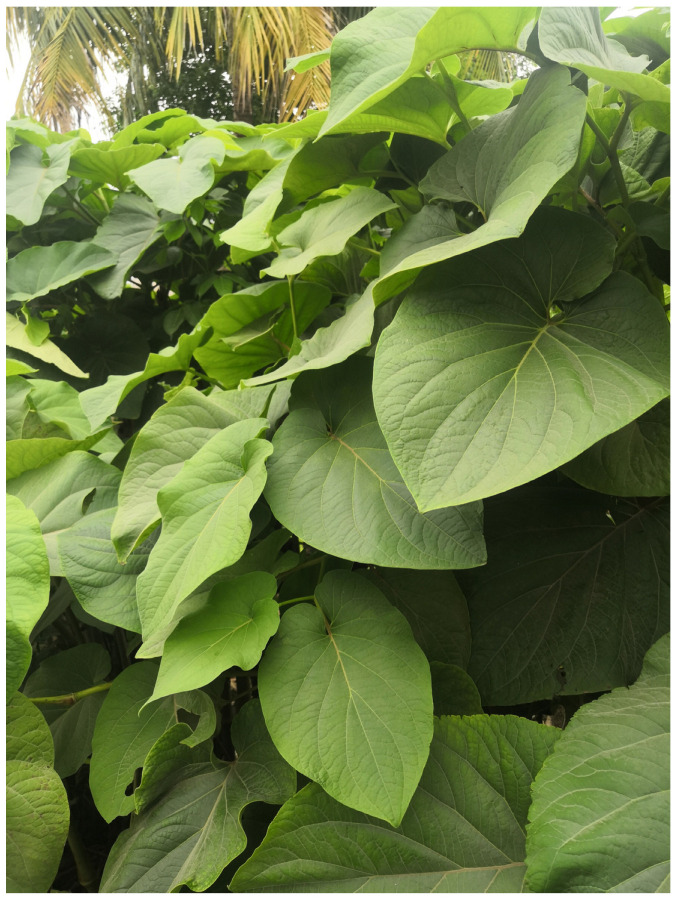
Fresh leaves of endemic *Piper cordoncillo* from the wild in Campeche, Mexico.

**Figure 2 insects-14-00312-f002:**
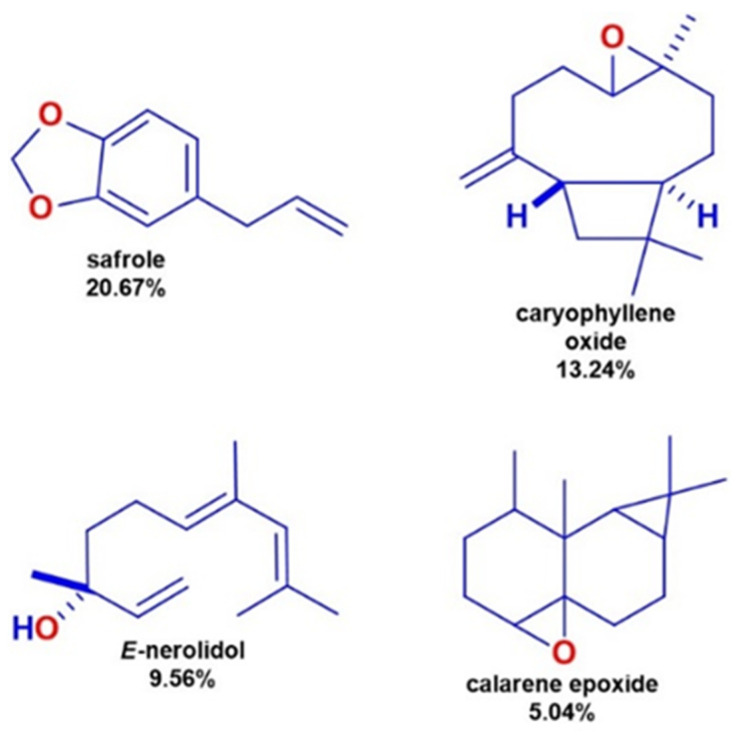
The major essential oil obtained from *Piper cordoncillo* essential oil.

**Figure 3 insects-14-00312-f003:**
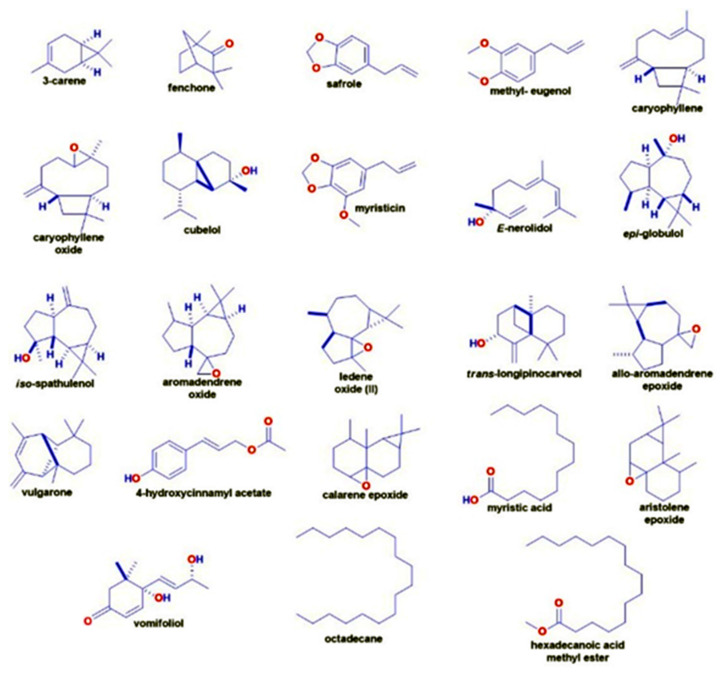
The chemical structure of essential oils present in *Piper cordoncillo* essential oil.

**Figure 4 insects-14-00312-f004:**
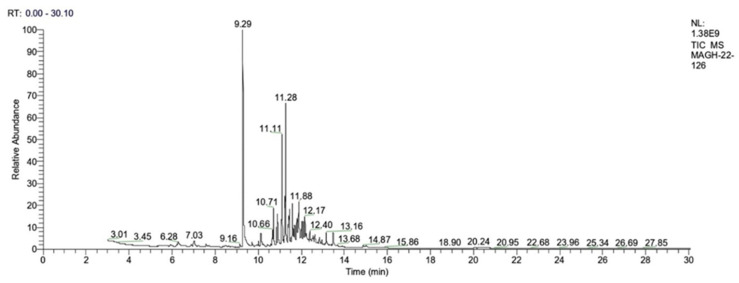
Chromatogram of *Piper cordoncillo* essential oil obtained by GC-MS.

**Table 1 insects-14-00312-t001:** Chemical composition of essential oil of *Piper cordoncillo* var. *apazoteanum*.

Peak	Compound	RT	Composition (%)	RRI Exp.	RRI Range ^1^
1	3-carene	6.28	0.78	1016	1002–1024 ^a^
2	Fenchone	7.03	1.25	1097	1078–1096 ^a^
3	Safrole	9.29	20.67	1268	1258–1289 ^a^
4	Methyl-eugenol	10.12	1.91	1372	1364–1402 ^a^
5	Caryophyllene	10.66	1.75	1428	1384–1430 ^a^
6	Cubelol	10.71	2.53	1490	1489 ^b^
7	*Epi*-globulol	10.85	1.02	1546	1540–1597 ^a^
8	Myristicin	10.89	4.30	1518	1503–1597 ^a^
9	*E*-Nerolidol	11.11	9.56	1540	1538–1565 ^a^
10	Caryophyllene oxide	11.28	13.24	1555	1549–1587 ^a^
11	Calarene epoxide	11.44	5.04	1672	1668–1671 ^g^
12	Allo-aromadendrene epoxide	11.59	3.87	1642	1646 ^d^
13	Ledene oxide II	11.69	1.51	1632	1636 ^c^
14	Aromadendrene oxide-(2)	11.79	2.17	1630	1630 ^c^
15	Trans-longipinocarveol	11.88	4.61	1633	1632 ^c^
16	Hexadecanoic acid methyl ester	12.01	1.4	1913	1902–1917 ^a^
17	Vulgarone	12.08	1.98	1648	1647 ^e^
18	C_15_H_26_O_2_	12.17	2.34	1677	-
19	*Iso*-spathulenol	12.24	1.13	1627	1619–1630 ^a^
20	Octadecane	12.4	1.08	1811	1800 ^j^
21	Myristic acid	12.54	1.07	1726	1720 ^h^
22	Aristolene epoxide	12.63	1.24	1752	1758 ^c^
23	Vomifoliol	12.84	0.86	1794	1796 ^i^
24	4-hydroxycinnamyl acetate	13.16	1.40	1666	1666 ^f^
TOTAL			86.71%		

^1^ RRI: relative retention indices; ^a^ Babushok et al. [50]; ^b^ Kalemba et al. [51]; ^c^ Abd-ElGawad et al. [52]; ^d^ Pattiram et al. [53]; ^e^ Skaltsa et al. [54]; ^f^ Ly et al. [55]; ^g^ Elshamy et al. [56]; ^h^ Alberts et al. [57]; ^i^ Lalel et al. [58]; ^j^ Chung et al. [59].

**Table 2 insects-14-00312-t002:** Cumulative mortality records of second instar larvae of *Aedes aegypti* mosquitoes treated with *Piper cordoncillo* essential oil for three consecutive days.

Concentration (ppm)	Total Number of Mosquito Larvae in 4 Replicates	Cumulative Mortality		
		N° Larvae per Day (h)		
		24	48	72
800	80	70.00 ± 8.16 a	95.00 ± 4.08 a	100.00 ± 0.01 a
400	80	46.25 ± 2.50 b	70.00 ± 4.08 b	85.00 ± 7.07 b
200	80	43.75 ± 4.79 b	56.25 ± 6.29 c	66.25 ± 4.79 c
100	80	38.75 ± 6.29 b	52.50 ± 6.45 c	62.50 ± 2.89 c
50	80	17.50 ± 6.45 c	25.00 ± 5.77 d	33.75 ± 4.79 d
Control	80	0.00 ± 0.00 d	0.00 ± 0.00 e	0.00 ± 0.00 e

Data represent the means of four replicates (*n* = 80). Means followed by different letters within the same columns and plant parts are significantly different at *p* < 0.05 compared with the control group.

**Table 3 insects-14-00312-t003:** Mortality (%) and relative growth index of the immature stages of *Aedes aegypti* mosquitoes treated with essential oil from *Piper cordoncillo*.

CONC (ppm)	Mortality (%)/N° de Larvae per Instar			Total	RGI Values	Category
	II	III	IV			
800	100.00 ± 0.01 a	-	-	100.00 ± 0.01 a	0.00 ± 0.00 a	++++
400	85.00 ± 7.07 b	5.00 ± 5.77 a	7.50 ± 2.89 ab	98.75 ± 2.50 a	0.05 ± 0.02 b	++++
200	66.25 ± 4.79 c	13.75 ± 4.79 a	11.25 ± 7.50 ab	91.25 ± 11.09 a	0.17 ± 0.04 c	++++
100	62.50 ± 2.89 c	12.50 ± 9.57 a	8.75 ± 7.50 ab	83.75 ± 14.93 ab	0.22 ± 0.06 cd	++++
50	33.75 ± 4.79 d	8.75 ± 8.54 a	17.75 ± 13.77 a	60.30 ± 20.60 b	0.45 ± 0.09 de	+++
Control	0.00 ± 0.00 e	0.00 ± 0.00 a	0.00 ± 0.00 b	0.00 ± 0.00 c	1.02 ± 0.01 e	-

Data represent means of four replicates (*n* = 4). Means followed by different letters (a, b, c, d, e) within the same column and plant part are significantly different at *p* < 0.05. CONC: Treatment concentration, RGI: Relative growth index; II, III, IV: Second, third, and fourth instar larvae; χ²(gL): Chi-squared (degrees of freedom); *p*-value: Probability value. Statistics= II: *p* < 0.001, F = 301.06, df = 5,18, χ² = 98.82, III: *p* = 0.060, F = 2.87, df = 4,15, χ² = 43.38, IV: *p* = 0.055, F = 2.95, df = 4,15, χ² = 44.03, Total: *p* < 0.001, F = 45.25, df = 5,18, χ² = 92.63, RGI: *p* < 0.001, F = 203.81, df = 5,18, χ² = 98.26. RGI-Category: (-) No inhibitory growth activity (RGI ≥ 1.00), (+++) High inhibitory growth activity (0.25 ≥ RGI ≥ 0.49), (++++) Very high inhibitory growth activity (0.00 ≥ RGI ≥ 0.24).

**Table 4 insects-14-00312-t004:** Probit linear regression analysis for mean lethal concentrations that control the population of *Aedes aegypti* mosquitoes treated with *Piper cordoncillo* essential oil for three consecutive days, and at the end of the 17-day experiment.

Days (h)	LC_50_ (ppm)	LC_90_ (ppm)	Slope (Standard Error)	χ² (df)	z-Valor	*p*-Valor
24	456.29 ± 37.74 (389.53–542.75)	1106.14 ± 101.40 (942.42–1360.59)	−0.8998 (0.088)	35.6293 (4)	−10.17	<0.001
48	230.30 ± 19.05 (193.63–269.73)	599.69 ± 43.10 (527.16–701.65)	−0.7990 (0.090)	42.1257 (4)	−8.81	<0.001
72	146.30 ± 12.62 (121.48–171.97)	379.52 ± 27.64 (333.30–445.63)	−0.8039 (0.1023)	44.6168 (4)	−7.86	<0.001
MT	61.84 ± 6.79 (47.74–74.93)	167.20 ± 11.49 (147.67–194.26)	−0.7523 (0.1221)	684.5260 (4)	−6.16	<0.001

MT: Total mortality (at the end of the 17-day experiment); LC: Lethal Concentration; LC_50_: Lethal concentration that controls half of the mosquito population; LC_90_: Lethal concentration that controls 90% of the mosquito population; χ² (df): Chi-squared (degrees of freedom); Z-value: Value of the statistical test (z-average); *p*-value: Probability value. Values in parentheses indicate the lower and upper confidence limits (95%).

## Data Availability

All the associated data are available in the manuscript.

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
