# Peer review of "Assessing the Larvicidal Properties of Endemic Campeche, Mexico Plant Piper cordoncillo var. apazoteanum (Piperaceae) against Aedes aegypti (Diptera: Culicidae) Mosquitoes"

_insects, 2023, doi:10.3390/insects14040312_

Round 1
Reviewer 1 Report
The draft of article number 2276316 submitted to the Insects, MDPI, entitled “Assessing the Larvicidal Properties of Endemic Campeche, Mexico Plant Piper cordoncillo var. apazoteanum (Piperaceae) against Aedes aegypti (Diptera: Culicidae) Mosquitoes” carried results in the text that needs some revision for the improvement of the draft. Some suggested changes for example are in the comments portion to revise and improve the manuscript. Please find suggested corrections, reference writing, journal-style format, author’s instructions, use of abbreviations and missing information for revision.
Line 55: eco-friendlier” Replace the word with some suitable word
Line 80, 82: P. marginatum and P. aduncum (Piperaceae” write the complete scientific names where first used. This may be applied throughout the whole manuscript
Mexico Plant Piper cordoncillo var. apazoteanum” please insert the pictures of the plant at suitable places if possible. Pictures of Momo
(w/v)” please write the detail of the abbreviation where first written
Line 139: P. cordoncillo” Italic
Line 141: EI mode: what is this abbreviation
Line 171: A. aegypti” italic
Line 190-192: RGI=Σ(No.de insectos vivos∗fase insecto)+ Σ[No.de insectos muertos∗(fase insecto−1)]4141(No.total de insectos evaluados∗total de fases del insecto)
The formula used may be written in the same language and the abbreviation used may be explained
Line 222: P. cordoncillo” italic
Please write the details of the abbreviations used in table 1.
Figure 2 may be shifted to supplementary figures, I would suggest please present the figure of those compounds obtained from the plant during the present study and the rest information shifted to the supplementary figures e.g figure 2 (if necessary)
Table 2: N° Larvae and (n = 80). “Please write with more clarity
Table 3: IRC Values: what are the abbreviations
Table 4: CL50 (ppm) and CL90 (ppm)” what is this” please use a similar format throughout the whole MS
Line 399: S. zeamais” complete scientific name where first used
Line 417-418: The study is the first to report on the biological activity and chemical composition of Piper cordoncillo var. apazoteanum, an endemic plant species from Campeche, Mexico.” Rephrase the sentence
Line 422: P. cordoncillo” italic
Please make a precise conclusion and also add future implications of the study
Please double-check for inconsistencies in Journal style/formatting/Reference writings/ authors instructions, double spaces, spellings of the words, English vocabulary, missing italics, scientific names, excessive/missing information, etc.
Author Response
Dear Reviewer,
We have incorporated the changes you suggested into the manuscript, following your precise recommendations. We are deeply grateful for your support in helping us improve our writing, and we will certainly take your suggestions into account in our future research.
Thank you for your valuable feedback.
Best regards,

Reviewer 2 Report
This manuscript focuses on the larvicidal properties of natural products for controlling A. aegypti by % mortality and growth inhibition, including identification of chemical compounds relating with insecticide activities. The content is interesting. However, there are some unclear points that I found in the manuscript.
1. Larvicidal bioassay: Why did the authors use the early second-instar A. aegypti larvae? Are they too small to use for detection and counting?
2. Why did you collect the data of mortality at 72 h after larvicidal testing? Was testing at 24 and 48 h enough for study effectiveness rate?
3. What is the importance of studying growth inhibition? More concentration of essential oils also showed higher inhibitory than less concentration. In the result of growth inhibition, please clarify how you apply this result on mosquito control.
4. Line 139, 171, 222, and 418 Please write italic letter in scientific name.
5. Line 191, Please change the formula to English version.
6. The content in table 4, please change to English version.
7. Please check the page number of ref 8 line 467-469 and ref 11 line 474-475.
Author Response

(The authors gave the same response as above.)

Reviewer 3 Report
See the attached file

Author Response

(The authors gave the same response as above.)

Round 2
Reviewer 3 Report
The paper was positively revised. However, I suggest to add an oxygen in C(1) to complete the chetone in the formula of vomifoliol (also named as belemenol A)
Author Response
Dear reviewer, we thank you in advance for your suggestions, which will undoubtedly improve our future work. We have made the suggested changes in the manuscript according to your observation. We have added an oxygen at C(1) to complete the chetone in the formula of Vomifoliol in Figure 3.
